# Comparative Study Effect of Urea-Sulfur Fertilizers on Nitrogen Uptake and Maize Productivity

**DOI:** 10.3390/plants11223020

**Published:** 2022-11-09

**Authors:** Samar Swify, Dovile Avizienyte, Romas Mazeika, Zita Braziene

**Affiliations:** 1Lithuanian Research Center for Agriculture and Forestry, Instituto al.1, LT-58344 Akademija, Lithuania; 2Soil and Water Department, Faculty of Agriculture, New Valley University, El-Kharga 72511, Egypt

**Keywords:** urea, sulfur, ammonium sulfate, urea cocrystal, nitrogen uptake, maize productivity

## Abstract

Combined nitrogen (N) and sulfur (S) fertilization is a good management strategy to reduce N loss and increase the efficiency of N fertilizers to achieve high grain yields and quality. Field trials for 2 yrs. (2018–2019) were conducted to evaluate the comparative advantage of conventional urea (150 N kg ha^−1^) compared to urea+ ammonium sulfate (150 N kg ha^−1^), urea+ calcium sulfate (150 N kg ha^−1^), and urea cocrystals (CaSO_4_.4urea) (150 N kg ha^−1^) when applied as nitrogen fertilizers to the maize. The statistics show a significant treatments effect on developed corn cobs, fresh and dry cob yields and grain yield, with 1000 grains with better results in 2019 than in 2018. The fertilization treatments affected grain yields significantly for 2018 and 2019, respectively. Urea+ ammonium sulfate and urea cocrystal provided a significant increase in grain yields by 10.5% and 7.50%, respectively, compared to urea in 2018, w1hereas, in 2019, urea cocrystal supplied the grain yields with a significant increase of 23.07% compared to urea, followed by urea + calcium sulfate which provided a 10.46% increase compared to urea. The study highlights that using urea-sulfur fertilizers enhanced the release of mineral nitrogen in the soil, improved the grain’s N uptake by the plant and increased maize grain yields.

## 1. Introduction

Recently, the use of urea nitrogen (N) fertilizers varies between conventional urea and modified-urea fertilizers. Due to the economic advantage of solid urea production compared to other nitrogenous fertilizers, urea has been increasingly used in some regions of the world since 1980 [1,2]. Moreover, urea is the only rich (45–46%) source of primary solid nitrogen and accounts for 35% of the world’s production [3]. Several nitrogen fertilizers were produced and developed, beginning with ammonium sulfate in 1923 and ending with sulfur-coated urea in the 1960s. However, N loss is still the main problem especially in urea fertilizers as some of the nitrogen may be lost as ammonia gas when applying urea on the soil surface layers [4,5], or as surface runoff, leaching to the ground and surface water [6].

Most fertilization research tries to reduce the nitrogen losses resulting from using urea as the primary nitrogen source and to improve its effectiveness for crop production [7,8,9,10,11]. Therefore, the best management practices for nitrogen fertilizers such as urea to maximize profit and reduce negative environmental impacts strongly depend on the input of fertilizers, which are mostly applied as a mixture of nutrients [12]. The use of N and S fertilizer mixing to meet crop requirements is a possible option to improve crop N and S use efficiencies, but this requires a good understanding of crop responses to N and S application.

Many studies report that urea use efficiency can be increased by mixing it with other materials such as sulfur [13,14,15,16,17,18,19,20] or organic acids [10,11,21,22,23,24,25,26,27,28,29,30], synthesizing and coating it with inert materials [21,31,32,33,34,35,36,37] or N stabilizers [2,8,38,39,40,41,42,43]. The coating delays the dissolving of N, while acid amendments lower the pH of the fertilizer strip, preventing or minimizing rapid hydrolysis and volatilization of ammonia, which may increase N use efficiency and reduce the potential for N losses [34,40,44].

Furthermore, various studies with a long series of fertilization trials have shown that N and S fertilization combinations showed positive interactions, greatly enhancing yield and quality in different crops [13,18,20,37,45,46,47,48]. Sulfur additions can improve nitrogen use efficiency (NUE) [13,14,15,16,18,20,48,49]. Integration of nitrogen (N) and sulfur (S) fertilization appears to be of particular importance and is among the successful strategies that have substantially enhanced the productivity of cereal crops [13,15,16,18,20,45,48]. Similarly, when S was applied at the highest N rate, N uptake increased, indicating harmony between both nutrients. Furthermore, concurrent N and S management is critical for lowering the potential contamination of residual soil nitrate by enhancing N recovery from the soil while maintaining high nitrogen use efficiency [35,50,51,52].

Thus, the main objectives of this study were (1) to evaluate the efficiency of conventional urea compared to urea-sulfur fertilization, (2) to reduce nitrogen losses and improve N uptake and (3) to improve corn’s yield production and grain quality. This study hypothesized that urea-sulfur fertilizers, especially synthesized urea (urea cocrystals), as modified urea fertilizers would significantly improve maize production because their slow release would meet the nutrient demands of maize during the whole growing period. Moreover, sulfur will improve nitrogen uptake by plants.

## 2. Materials and Methods

### 2.1. Experiment Location and Soil Characteristics

Field trials for 2 years (2018–2019) were carried out at Rumokai Experimental Station of the Lithuanian Center for Agriculture and Forest Sciences (54°43′15.7044′′ N, 22°58′36.667′′ E). The soil was Hapli-Epihypogleyic Luvisol (LVg-p-w-ha) [53] with a moderately heavy loam texture. The soil chemical properties at 0–20 cm, except for nitrogen form concentrations and available sulfur at the soil surface layer (0–30) cm, are shown in Table 1.

### 2.2. Research Schema and Experimental Design

The field trials for 2 years (2018–2019) were laid out in a randomized complete block design (RCBD) with 20 experimental plots. Five treatments were arranged in four replicates with a total plot area of 4 m^2^ (2 m × 2 m). The treatments used, consisted of:Control (N0P80K160 kg ha^−1^).Urea (N150P80K160 kg ha^−1^).Urea + ammonium sulfate (N150P80K160S42.5 kg ha^−1^).Urea + calcium sulfate (N150P80K160S42.5Ca53 kg ha^−1^).Urea Cocrystal as CaSO_4_.4urea (N150 P80K160S42.5Ca53 kg ha^−1^).

### 2.3. Field Preparation and Maize Cultivation

The field preparation and maize cultivation dates during 2018 and 2019 are shown in Table 2. Maize (Ramirez, characterized as FAO 160) seeds were sown manually one day after fertilization. Row spacing was 50 cm, plant spacing was 20 cm and maize density was 10 plants per square meter (100,000 plants ha^−1^). Pesticides were not used. Weeds were controlled manually as needed and plants were grown under arable farming conditions.

### 2.4. Soil Sampling and Analytical Procedures

The soil samples were taken at 0–30 cm depth from the non-treated and treated plots. A stainless-steel push probe was used to take the samples with three subsamples per plot composited to make one sample. The samples were air-dried and ground to pass through a 2 mm sieve. All the soil properties, mineral nitrogen and sulfur analyzes were performed in the Agrochemical Research Laboratory at Lithuanian Research Center for Agriculture and Forestry. Soil pH was determined using a 1:5 *(v/v*) soil suspension in the 1 M KCl (ISO 10390:2005) [54]. Soil-available phosphorus as P_2_O_5_ and potassium as K_2_O were extracted using 1:20 (*wt./v*) soil suspension of ammonium lactate-acetic acid extractant (pH 3.7) [54]. Soil available P_2_O_5_ was determined using ammonium molybdate via the spectrometric method with a Shimadzu UV 1800 spectrophotometer (LVP D-07:2016). Mobile K_2_O was determined using flame emission spectroscopy with a flame emission spectroscopy JENWAY PFP7 flame photometer (LVP D-07:2016).

Soil available sulfur was determined by laboratory-prepared the LVP D-12–2011 turbidimetric method. Mineral nitrogen was determined by using a spectrometric flow injection analysis (FIA) method developed by the laboratory; nitrate content (Sum of N^_^NO_3_ and N^_^NO_2_; LVP D-05:2016) and ammonium content (N^_^NH_4_; LVP D-05:2016) were determined. The mineral nitrogen is calculated by adding the sum of nitrate and nitrite nitrogen to ammonium nitrogen. The organic soil carbon content was determined using dry combustion according to ISO 10694:1995, where the sample was heated to 900 °C in a stream of air and the carbon dioxide formed was measured using infrared spectroscopy. To evaluate plant biometric parameters, only the inner portion of the plots (2 × 2 m) was harvested at the physiological maturity stage. Twenty plants were randomly selected for grain yield and grain biomass quality determination. Samples were taken from all the replicates and oven-dried at 65 ± 5 °C until constant weight to obtain dry biomass and yield weight.

### 2.5. Nitrogen Uptake and Its Efficiency

The fertilizer N uptake and apparent nutrient recovery efficiency (ANR) were calculated by the following formulas [15,55]:(1)$$N\;\mathrm{Uptake}=\%N\;\mathrm{in}\;\mathrm{grains}\times \mathrm{dry}\;\mathrm{matter}\;\mathrm{of}\;\mathrm{grains}\;\left({\mathrm{kg}\;\mathrm{ha}}^{-1}\right)$$

Apparent nitrogen recovery efficiency (ANR) has been used to reflect a plant’s ability to acquire applied nutrients from soil [56]:(2)$$\%\;\mathrm{ANR}=\frac{\mathrm{Upake}\;F,\;\mathrm{kg}-\mathrm{Uptake}\;C,\;\mathrm{Kg}}{\mathrm{Quantity}\;\mathrm{of}\;\mathrm{fertilizer}\;\mathrm{applied},\;\mathrm{kg}}\times 100\;$$

### 2.6. Climatic Conditions

Considerable variations were observed in seasonal climatic conditions between 2018 and 2019, as shown in Figure 1. In 2018, the air temperature was warmer than average in 2019, except in June. As a result, harvesting occurred 1 month earlier that year (2018). On the other hand, the 2019 season was less rainy compared to 2018, especially in April and June, which affected delayed corn cultivation and early crop development.

### 2.7. Statistical Analysis

Analysis of variance (ANOVA) was performed using a general linear model on plants’ density, total green matter, dry matter yield and grain yields. Grain characteristics (developed corn cobs, fresh cobs yields and dry cobs yields), N uptake and ANR. Person’s correlation analysis was performed to determine the relationship between time and soil mineral N and its forms (nitrate and ammonium) and soil available sulfur content and grains total N content and N uptake. The statistical analysis software was IBM SPSS 25.0. and Duncan’s test at the 5% level was performed to separate means according to ANOVA results.

## 3. Results

### 3.1. Nitrogen Release in Soil-Grown Maize from Urea-Sulfur Fertilizers

To investigate the influence of urea fertilizer on maize productivity and nitrogen accumulation and its uptake with sulfur fertilizers use, soil-grown maize plants were supplied with U (urea), UAS (urea + ammonium sulfate), UCS (urea + calcium sulfate) and UCSC (urea cocrystals–CaSO_4_.4urea). The mineral nitrogen in the soil formed into nitrate (N-NO_3_) and ammonium (N-NH_4_) as shown in Figure 2. The results indicate that the concentration of nitrate N-NO_3_ and ammonium N-NH_4_ in the soil was higher in 2019 than in 2018.

The concentration of nitrate N-NO_3_ in the soil in both years (2018 and 2019) correlated significantly (*p* < 0.000 and *p* < 0.001) with the time during the maize growth period (r = −0.66 and r = −0.46) in 2018 and 2019, respectively. The nitrate started with a low concentration before fertilization with means of 10.11 and 11.82 mg kg^−1^ in 2018 and 2019, respectively (Figure 2A,B).

After one week, in both years (2018–2019), the fertilizer treatment of UCS recorded the highest values of nitrate concentration in the soil with a mean of 32.75 and 34.46 mg kg^−1^, respectively (Figure 2A,B), whereas, UCSC showed the lowest value of nitrate in both years, with a mean of 24.48 and 26.19 mg kg^−1^, respectively, after the control. In 2018, nitrate concentrations correlated significantly with treatments. During the period from 2–10 weeks, concentration flow fluctuated from increase to decrease for all the treatments (Figure 2A).

By the end of the tenth week in 2018, the treatments of UAS > U > UCSC showed high values of nitrate with means of 16.13 > 15.16 > 14.30 mg kg^−1^, respectively (Figure 2A). After 10 weeks, the nitrate concentration tended to decrease till the harvest. After the harvest (AH), the treatments of UCS > UAS > UCSC recorded high nitrate concentration in the soil with means of 6.93 > 4.39 > 3.88 mg kg^−1^, respectively. In 2019, the nitrate concentration tended to increase for all the treatments after fertilization till 4 weeks.

After 4 weeks, the curve showed a decrease for all the treatments until the end of the maize growing period (Figure 2B). In addition, the treatment of UAS recorded the highest values of nitrate concentration in the soil during the whole period of maize growth (Figure 2B), then followed by the treatment of UCSC. The mean of UAS and UCSC nitrate concentration ranged from 11.82 mg kg^−1^ before fertilization to 44.29 and 38.67 mg kg^−1^, respectively, after 4 weeks. Moreover, UAS recorded the highest nitrate concentration at 13.48 mg kg^−1^ after harvest (AH), as shown in Figure 2B. However, in the 2019 season the concentration of nitrate was not correlated significantly with the time for all the fertilization treatments except for the treatment of UCSC (r = −0.71*).

Furthermore, the results showed the concentration of ammonium in the soil was low in 2018 and 2019 compared to nitrate, as shown in Figure 2C,D. It represents approximately 9 and 17% of the total mineral nitrogen in the soil in 2018 and 2019, respectively. The ammonium concentration was not correlated significantly with the time or with the fertilization treatments in both years (2018–2019). The ammonium concentration started with means of 1.81 and 1.86 mg kg^−1^ in 2018 and 2019, respectively. In 2018, urea showed the highest value of ammonium with an average of 2.29 mg kg^−1^, while in 2019 the treatment of UAS recorded the highest value of ammonium concentration (Figure 2D) followed by urea with averages of 3.85 and 2.97 mg kg^−1^, respectively. After the harvest, the U and UCSC recorded high ammonium concentrations with means of 2.37 and 2.28 mg kg^−1^ in 2018 and 3.34 and 3.47 mg kg^−1^ in 2019, respectively (Figure 2C,D).

Figure 3A,B show the mineral nitrogen concentration in the soil during the maize growth period in 2018–2019. The soil mineral N concentration in both years (2018–2019) correlated significantly (*p* > 0.000 and *p* > 0.001) with time. The soil mineral N showed a negative linear relationship (r = −0.65 and r = −0.45) with time which means the concentration decreased over time. Before fertilization, soil mineral nitrogen was low with a mean of 11.92 and 13.68 mg kg^−1^ in 2018 and 2019, respectively.

In addition, the mineral nitrogen for the fertilizers’ treatments correlated significantly with time in 2018, while in 2019 the concentration was not correlated significantly for all fertilization treatments except UCSC (r = −0.69*). In 2018, during the first 2 weeks, the treatments of UCS and UAS had the highest mineral N concentration values, followed by U and UCSC (Figure 3A). After 2 weeks, UCSC showed the highest mineral N concentration till the maize harvest (Figure 3A).

In 2019, the UAS had the highest mineral N concentration during the whole of the maize growing season (Figure 3B). The highest residual values of the mineral nitrogen after the maize harvest were observed in the treatment of UAS with 6.54 and 16.63 mg kg^−1^ in 2018 and 2019, respectively. The concentration of soil mineral N and its forms N-NO_3_ and N-NH_4_ correlated significantly (*p* < 0.000) with soil sulfur in 2019, but no significant correlation between the sulfur concentration in soil and the mineral N and nitrate was reported in 2018. However, ammonium correlated significantly (*p* < 0.000) with the sulfur in the soil (r = 0.52). UAS recorded the highest sulfur concentration during both years (2018–2019), followed by UCSC and UCS (Figure 3C,D).

### 3.2. Plants’ Density, Green Matter and Dry Matter Yield

Table 3 shows the mean of the plants’ density and green and dry matter yields after using conventional urea compared to urea-sulfur fertilizers. As was expected, in both years (2018–2019) the treatments were significantly higher than those obtained in the control. In 2018, the treatments significantly affected the plants’ density and green matter yields compared to the control and urea + ammonium sulfate (UAS) recorded a high plant density and green matter yield (Table 3), while no significant effect was observed on plants’ density and green matter yields in 2019. However, urea cocrystals (UCSC) recorded a significant density and green matter yield compared to urea, as shown in Table 3. For both years (2018- 2019), the treatments had no significant effects on dry matter yields, but UCSC recorded the highest dry matter yields with 8.43 and 11.41 t ha^−1^ for both seasons 2018- 2019, respectively, as shown in Table 3.

### 3.3. Grain Yields and Grain Quality

ANOVA showed significant treatment effects on developed corn cobs, fresh and dry cob yields and grain yields, and 1000 grains. The extent of treatments‘ impact on corn productivity differed across the two study years (2018–2019) as shown in Table 4, with better results in 2019 than in 2018.

Generally, the maize with UAS and UCSC always had a significantly higher effect for all productivity parameters during both years. (2018–2019). In 2018, the maize with UAS recorded the significantly highest fresh and dry cob and grains yields and 1000 grains, while in 2019, UCSC was significantly the highest for all productivity parameters, as shown in Table 4. UCSC also recorded the highest values for the developed corn cobs’ yield in both years (2018–2019).

The grain yields were affected significantly for both years (2018 and 2019), as shown in Figure 4, but there were no significant differences observed when the yield was compared in both seasons (2018–2019). In 2018, UAS and UCSC provided a significant increase in grain yields by 47.8% and 43.73%, respectively, higher than the control and 10.5% and 7.50%, respectively, higher than urea; whereas, in 2019, UCSC supplied the grain yields with a significant increase, 41.17% and 23.07% higher than the control and urea, respectively, followed by UCS, which provided a 26.70% and 10.46% increase higher than the control and urea, as shown in Table 4. The treatment of UAS was in the third rank by 23.52 and provided a 7.60% increase, higher than the control and urea, respectively. The 1000 grains weight was not affected by the treatments in 2018, but there was a significant effect in 2019 and UCSC recorded the highest weight with 286.02 g for 1000 grains as shown in Table 4.

Grain quality characteristics including the total nitrogen and crude protein and starch contents are listed in Table 5. The fertilization treatments significantly affected the grain’s quality characteristics including the total nitrogen and crude protein contents, except that starch was affected significantly in 2019 only (Table 5).

In both years (2018 and 2019), UCSC recorded the highest total nitrogen content in grains by 1.42 and 1.47%, respectively. Additionally, UCSC showed the highest crude protein in grains by 8.87 and 9.22% in 2018 and 2019, respectively as shown in Table 5. The starch content had no significant effect by fertilizer treatments in 2018 but was affected significantly (*p* < 0.000) in 2019, and urea treatment showed the highest starch content at 74.99% followed by the control with 74.64%. In addition, the grain’s sulfur content was affected significantly (*p* < 0.000) and the treatment of UAS recorded the highest mean average for both years. (2018–2019) by 975 mg kg^−1^.

In 2018, soil mineral nitrogen and sulfur correlated significantly (*p* < 0.000) with the total nitrogen and the crude protein in the grains (Table 5). The correlation coefficients (r = 0.94 and r = 0.93) showed a strong positive linear relationship between soil mineral nitrogen and total nitrogen and the crude protein content in the grains, respectively. Moreover, sulfur (r = 0.79) showed a positive linear relationship between total nitrogen and the crude protein content in the grains (Table 5). Soil mineral nitrogen and sulfur also correlated significantly *p* < 0.05 and *p* < 0.000) with the grain’s total nitrogen and the crude protein in 2019. A medium positive linear relationship between soil nitrogen and grain’s total nitrogen and the crude protein content is shown by the correlation coefficient r = 0.41 and r = 0.47, respectively.

In addition, sulfur (r = 0.61 and r = 0.58) showed a positive linear relationship between total nitrogen and the crude protein content in the grains, respectively. The soil mineral N and sulfur had no significant correlation with the grain’s starch content in both years. (2018–2019). Additionally, the grain’s total nitrogen correlated significantly (*p* < 0.000) with the crude protein and starch content in the grain (Table 5) in both years (2018–2019).

The correlation coefficients (r = 0.99 and 0.98) showed a positive strong linear relationship between the total nitrogen and crude protein content in the grains in both years. (2018–2019), as shown in Figure 5A,B, while the starch content showed a significant correlation (*p* < 0.000) with the grain’s total N but with a negative linear relationship (r = −0.77 and r = −0.39) in both years (2018–2019), as shown in Figure 5A,B.

### 3.4. Nitrogen Uptake and Apparent Nitrogen Recovery Efficiency in Grains

Analysis of variance for grain N uptake and apparent nitrogen recovery (ANR) values are listed in Table 6. The N uptake and ANR were affected significantly (*p* < 0.000) by the fertilizer treatments compared to the control in the 2 years (2018–2019). In 2018, there were no significant differences in N uptake and ANR between the fertilizers’ treatments but the treatment of UAS recorded the highest N uptake by 142.45 kg ha^−1^ and the highest nitrogen recovery value in the grains with 43.44%, respectively.

In 2019, the statistics showed significant differences between the fertilizers’ treatments in N uptake and ANR as shown in Table 6. The treatment of UCSC recorded the highest N uptake and ANR with 173.31 kg ha^−1^ and 37.34%, respectively. The soil mineral N correlated significantly (*p* < 0.000 and *p* < 0.001) in 2018–2019 with grain N uptake (Table 6). As a result of increasing the grain’s N uptake, the grain’s total N (*p* < 0.007 and *p* < 0.49) and grain yields increased significantly (*p* < 0.016 and *p* < 0.000) in 2018 and 2019, respectively. In 2018, the correlation coefficient provided a strong linear relationship (Figure 6) between the soil mineral N and the N uptake in the grains (r = 0.79**).

In 2019 the correlation coefficient also provided a medium linear relationship between the soil mineral N and the N uptake in the grains (r = 0.62**) as shown in Figure 7, moreover providing a strong linear relationship between the N uptake and the grains yield (r = 0.99** and r = 0.96**) in 2018 and 2019, respectively. In addition, soil available sulfur correlated highly significantly in both years. (2018 and 2019) with grains N uptake (*p* < 0.003 and *p* < 0.000), with a medium linear relationship as shown in Figure 6 and Figure 7.

## 4. Discussion

The extent to which treatments impacted corn productivity differed between the two study years (2018–2019), with 2019 showing better results than 2018. Generally, maize with UAS and UCSC had a significant effect on all productivity parameters during both years (2018–2019).

### 4.1. Effect of Sulfur Fertilizers on Mineral N Release and Nitrogen Uptake

Current results revealed a significant effect of participatory application of urea-sulfur fertilizers on the concentration of soil mineral N compared to conventional urea only. As was expected, the increase of sulfur in the soil enhanced significantly the release of soil mineral N from the urea granules [16,17,57].

The mineral nitrogen in the soil formed into nitrate (N-NO_3_) and ammonium (N-NH_4_) as shown in Figure 2. The results indicate that the concentration of nitrate N-NO_3_ and ammonium N-NH_4_ in the soil was significantly higher in 2019 than in 2018. Many factors affect the release process of nitrogen from the urea granules [58,59,60]. The meteorological conditions during the maize cultivation period played a vital role in the release of soil mineral N. The environmental variables, especially rainfall and temperature over the study period, had a high effect on the release of N and N uptake [58,59,61,62], which showed that the 2019 season was less rainy compared to 2018, and colder (Figure 1). This resulted in a slower release rate for the fertilizer treatments, less ammonia loss, and a lower nitrate leaching, which means a higher accumulation of nitrate and ammonium in the soil [59,63,64].

By detecting the role of sulfur fertilizers in enhancing the release of soil mineral N, the concentration of mineral N and its forms N-NO_3_ and N-NH_4_ correlated significantly in 2019 but with no significant correlation between the sulfur concentration in soil and the mineral N and nitrate in 2018. However, ammonium correlated significantly with sulfur in the soil. It is important to note that, while equivalent in performance in urea-sulfur fertilizers, S sources are different in their solubility. Therefore, the treatments were different in sulfur release [34,65]. UAS recorded the highest sulfur concentration during both years. (2018–2018), followed by UCSC and UCS (Figure 3C,D). The exact solubility of S in CaSO_4_.4urea is not known but can be suggested to follow that of CaSO_4_ or CaSO_4_·2H_2_O, while (NH_4_)_2_SO_4_ is highly soluble and thus prone to nutrient loss. Hence, not only is it available to the crops but can potentially be available for the plants for a longer time in the environment [38,66,67,68,69]. This result is supported by several previous studies that reported that suitable S source application promotes the absorption of N in the soil [18,19,20,70].

As a result, increasing the concentration of soil mineral N enhanced the N uptake of the grains [18,19,20,57]. Maize N uptake of grain was significantly greater in 2019 than in 2018. The fertilizer treatments had significant effects on grain N uptake and ANR compared to the control in both years. (2018–2019). There were no significant differences in N uptake and ANR between the fertilizer treatments in 2018, but the treatment of UAS recorded the highest N uptake by 142.45 kg ha^−1^ and the highest nitrogen recovery value in the grains with 43.44%, respectively. The statistics showed significant differences between fertilizer treatments in N uptake and ANR (Table 6) in 2019. The treatment of UCSC recorded the highest N uptake and ANR with 173.31 kg ha^−1^ and 37.34%, respectively. The soil mineral N correlated significantly (*p* < 0.000 and *p* < 0.001) in 2018–2019 with grain N uptake (Table 6).

### 4.2. Effect of Combining Urea and Sulfur on Grain Yields and Grain Quality

The present study on maize crops demonstrated the positive effect of urea-N with S fertilization on grain yield. Nutrient management was an important strategy to increase maize yield and grain protein concentration [9,37,71,72,73]. In this study, the combination of urea and sulfur fertilizers could sustainably support the increase of accumulation of soil mineral N over time and increase the N uptake in grains, which means an increase in the grain’s protein concentration [15,18,19,20]. Conventional nitrogen fertilization alone such as with urea could increase grain yield, but the urea-N losses were higher compared to urea with ammonium sulfate or calcium sulfate [15,31,42,67,74,75,76,77].

This resulted in less grain yield and less quality and characteristics of grains for both years. (2018–2019), as shown in Table 5. Nitrogen assimilation is linked to S-metabolism, so as S metabolism slows, so does N assimilation if the S supply is insufficient [59]. Sulfur is reported to improve the photosynthetic assimilation of N in proteins at the expense of nonproteins in crops [57,78], which increases dry matter yield, as 90% of plant dry weight is thought to be derived from photosynthesis products [19,78]. In this study, the significant increase in grain yield and ANR as a result of fertilization of urea with sulfur further showed a link of N with S. The increase in grain N concentration and content of protein was significant for both years. (2018 and 2019).

Response to N and S fertilization was mainly observed in maximizing grain yield, especially in 2019. The fertilization of urea + ammonium sulfate and urea cocrystal and urea + calcium sulfate provided a significant increase in grain yields, higher than the control and urea, in 2018 and 2019 as shown in Table 4, because of the high content of these treatments of sulfur compared to the treatments of urea and control [20,79,80].

In addition, the urea-sulfur treatments significantly affected the grain’s quality characteristics including the total nitrogen and crude protein and sulfur contents. In both years. (2018 and 2019), urea cocrystals recorded the highest total nitrogen content in grains by 1.42 and 1.47%, respectively. In addition, urea cocrystals showed the highest crude protein in the grains by 8.87 and 9.22% in 2018 and 2019, respectively, as shown in Table 5. Sulfur fertilization significantly increased the S concentration in grains [57]. The highest sulfur concentration was observed in the treatment of urea + ammonium sulfate due to its high solubility compared to urea+ calcium sulfate and urea cocrystal. Our results showed the same strong interaction between N and S as reported in many previous studies [18,19,20,70].

## 5. Conclusions

The use of N and S as urea-sulfur fertilizers had a significant impact on maize yield, characteristics and grain quality. The participatory regulation of N and S considerably enhanced maize N uptake and nutritional quality, meeting the requirements for sustainable development in maize production and providing a new theoretical basis and method for high-yield and high-quality maize production. This study reported that a positive trend of mineral sulfur-nitrogen (MSN) accumulation in soil over time increased N uptake, especially with the synthesized urea N source. This could reduce the potential for environmental pollution by nitrate leaching and ammonia volatilization. Appropriate sources of N such as urea mixed with sulfur additives could optimize crop nitrogen use efficiency.

## Figures and Tables

**Figure 1 plants-11-03020-f001:**
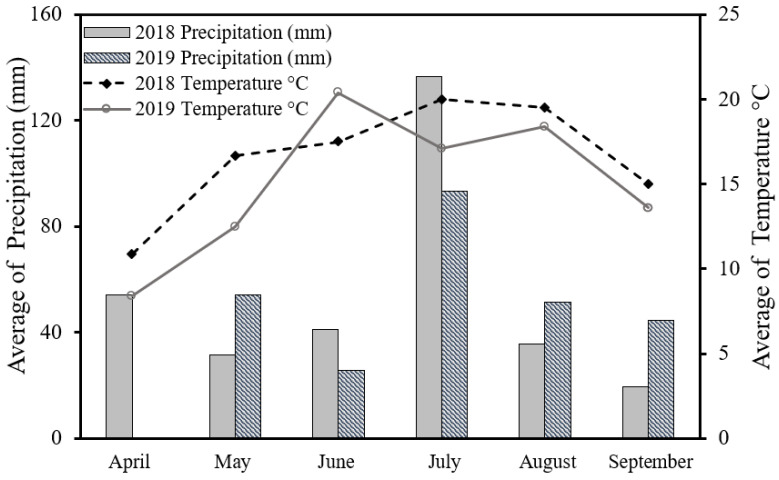
Mean monthly temperatures °C and precipitation (mm) during the growing maize season for the 2 years study (2018–2019).

**Figure 2 plants-11-03020-f002:**
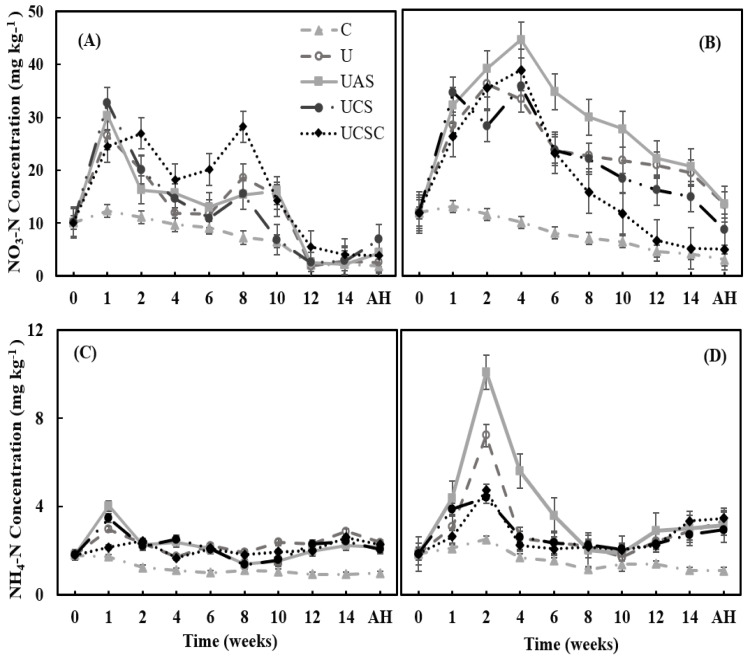
The release of nitrate NO_3_-N (**A**,**B**) and ammonium NH_4_-N (**C**,**D**) with time from the fertilizer treatments in soil surface layer 0–30 cm during the maize growth period and after harvest (AH) in 2018 (**A**,**C**) and 2019 (**B**,**D**). C = Control U = Urea, UAS = Urea + ammonium sulfate, UCS = Urea + CaSO_4,_ and UCSC = Urea cocrystal (CaSO_4_.4urea). (**A**) C r = −0.956**; U r = −0.715*; UAS r = −0.705*; UCS r = −0.714* and UCSC r = −0.654*; (**B**) C r = −0.986**; U r = −0.455; UAS r = −0.433; UCS r = −0.616 and UCSC r = −0.713*; (**C**) C r = −0.802**; U r = 0.267; UAS r = −0.359; UCS r = −0.259 and UCSC r = 0.350; (**D**) C r = −0.834**; U r = −0.212; UAS r = −0.398; UCS r = −0.273 and UCSC r = 0.103. * Significant differences at *p* < 0.05; ** Significant differences at *p* < 0.01.

**Figure 3 plants-11-03020-f003:**
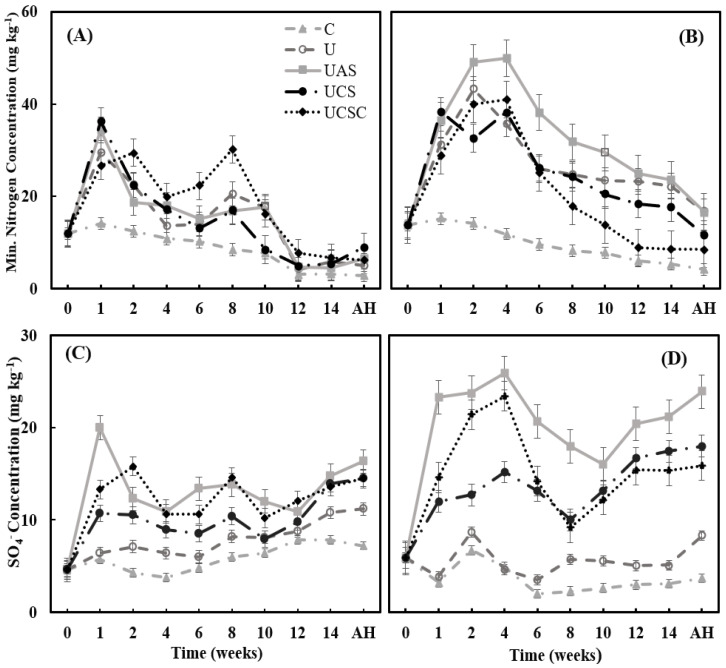
The release of mineral nitrogen (**A**,**B**) and soil available sulfur (**C**,**D**) with time from the fertilizer treatments in soil surface layer 0–30 cm during the maize growth period and after harvest (AH) in 2018 (**A**,**C**) and 2019 (**B**,**D**). C = Control U = Urea, UAS = Urea + ammonium sulfate, UCS = Urea + CaSO_4,_ and UCSC = Urea cocrystal (CaSO_4_.4urea). (**A**) C r = −0.964**; U r = −0.698*; UAS r = −0.699*; UCS r = −0.704* and UCSC r = −0.648*; (**B**) C r = −0.981**; U r = −0.441; UAS r = −0.453; UCS r = −0.611 and UCSC r = −0.694*; (**C**) C r = 0.843**; U r = 0.927**; UAS r = 0.279; UCS r = 0.670* and UCSC r = 0.380; (**D**) C r = −0.528; U r = 0.182; UAS r = 0.253; UCS r = 0.759* and UCSC r = 0.034; * Significant differences at *p* < 0.05; ** Significant differences at *p* < 0.01.

**Figure 4 plants-11-03020-f004:**
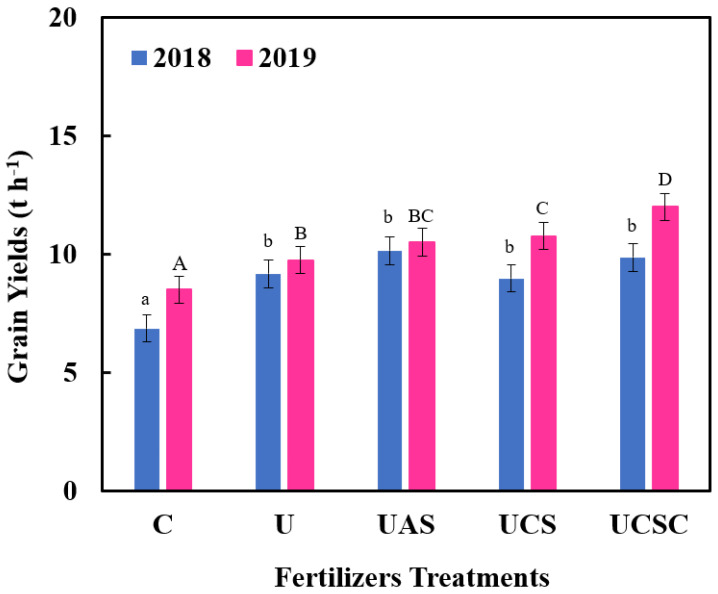
The effect of the urea-sulfur fertilizers on grain yields for 2 years. (2018–2019). Control U = Urea, UAS = Urea + ammonium sulfate, UCS = Urea + CaSO_4_ and UCSC = Urea cocrystal (CaSO_4_.4urea). Note 1. Columns followed by the same letter with the same size are not different (*p* < 0.05) according to Duncan’s multiple range test at the 5% level. Note 2. No significant differences between both seasons.

**Figure 5 plants-11-03020-f005:**
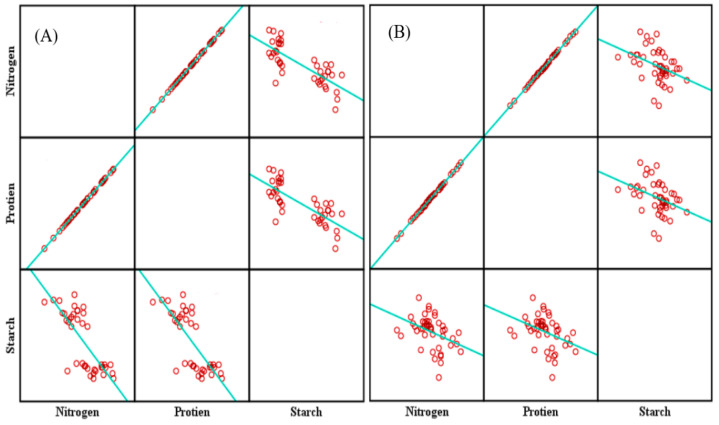
The matrix correlation between the total nitrogen (%) and the crude protein (%) and starch content (%) in the maize grains during 2018 (**A**) and 2019 (**B**).

**Figure 6 plants-11-03020-f006:**
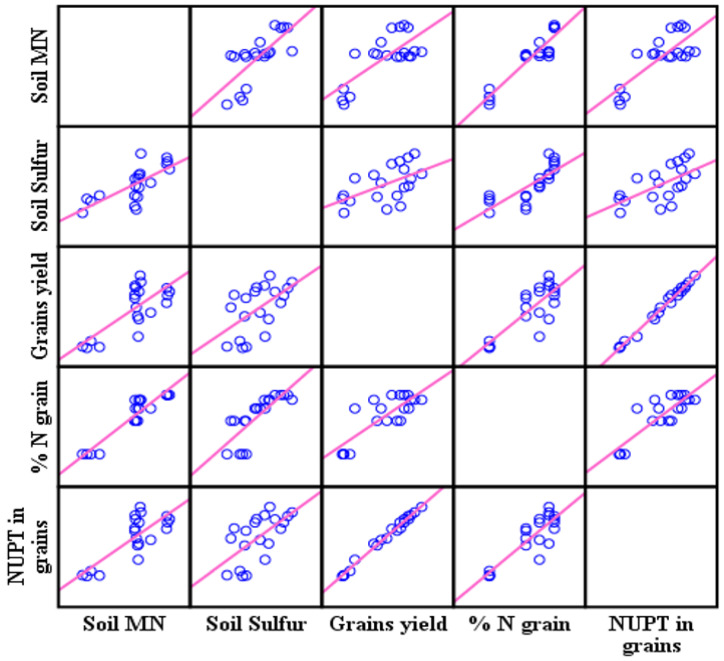
The correlation matrix relations effect of soil mineral N and sulfur on the N uptake in the grains, the grain’s total N, and the grain’s yield in 2018.

**Figure 7 plants-11-03020-f007:**
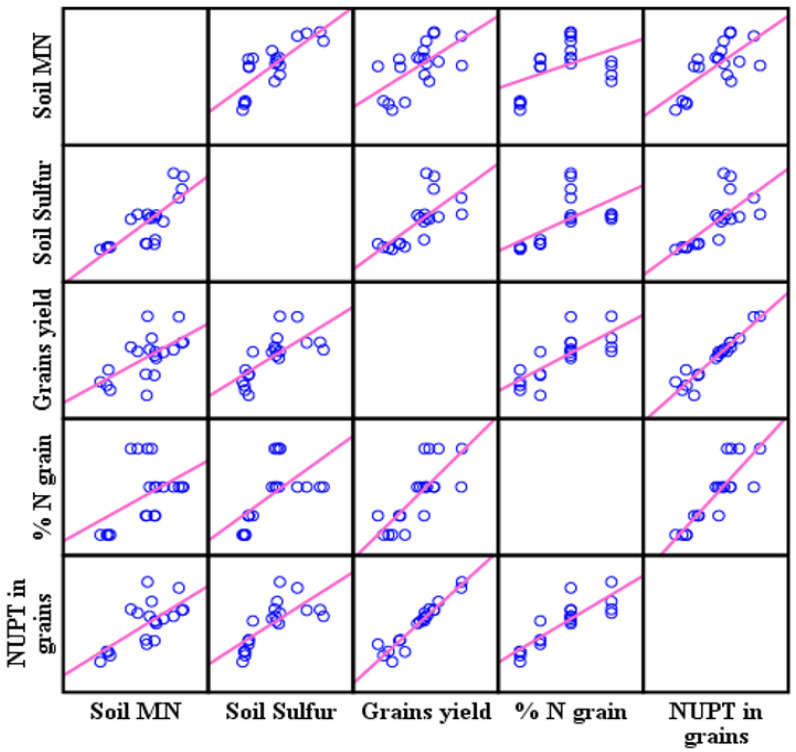
The correlation matrix relations effect of soil mineral N and sulfur on the N uptake in the grains, the grain’s total N, and the grain yield in 2019.

**Table 1 plants-11-03020-t001:** Soil chemical characteristics before using fertilizers in 2018 and 2019.

Soil Properties	pH _KCL_	SOC *	P_2_O_5_	K_2_O	S-SO_4_ **	Nitrogen (mg kg^−1^)		
	-	%	mg kg^−1^			Min. N	NO_3_-N	NH_4_-N
Depth (cm)	0–20				0–30			
2018	6.5	1.31	165	182	4.6	11.92	10.11	1.81
2019	7.4	1.41	249	198	5.9	13.68	11.82	1.86

* Soil organic carbon; ** Soil available sulfur.

**Table 2 plants-11-03020-t002:** Field preparation and maize cultivation dates in 2018 and 2019.

Process	2018	2019
Plowing (20–22 cm)	25 October 2017	27 October 2018
Surface tillage	25 April 2018	29 April 2019
Fertilization	2 May 2018	6 May 2019
Maize sowing	3 May 2018	7 May 2019
Harvest *	23 August 2018	20 September 2019

* The harvest was at the physiological maturity stage.

**Table 3 plants-11-03020-t003:** Means of plants’ density, total fresh and dry matter yields after using urea-sulfur fertilizers for the 2-year study (2018–2019).

Treatments	Plants’ Density	Green Matter	Dry Matter	% Moisture
	1000 Plant ha^−1^	t ha^−1^		
2018				
Control	92.50 a	17.90 a	6.29	64.86
Urea	97.50 b	20.00 b	7.52	62.40
Urea + (NH_4_)_2_SO_4_	100.00 b	22.33 c	7.59	66.00
Urea + CaSO_4_	98.75 b	21.88 c	7.84	64.16
CaSO_4_.4urea	97.50 b	22.05 c	8.43	61.77
SE±	1.21	0.38	0.53	-
*p*-value	0.009	0.000	ns *	-
2019				
Control	92.83 a	17.47a	8.92	48.94
Urea	93.98 a	18.93a	9.83	48.07
Urea + (NH_4_)_2_SO_4_	97.60 ab	18.79a	9.85	47.58
Urea + CaSO_4_	95.55 ab	18.22a	9.66	46.98
CaSO_4_.4urea	100.00 b	22.12b	11.41	48.42
SE±	1.59	0.85	0.92	-
*p*-value	0.049	0.020	ns *	-

Note. values in the same column followed by the same letter are not different (*p* < 0.05) according to Duncan’s multiple range test at the 5% level. * ns = not significant.

**Table 4 plants-11-03020-t004:** Means of developed corn cobs’ number, fresh and dry cob yields, grain yields, and 1000 grains after using urea-sulfur fertilizers for the 2 years study (2018–2019).

Treatments	Developed Corn cobs	Fresh Cobs Yields	Dry Cobs Yields	Grain Yields	1000 Grain
	1000 ha^−1^	t ha^−1^			g.
2018					
Control	80.00 a	12.14 a	7.82 a	6.86 a	222.52
Urea	93.75 b	17.39 b	11.34 b	9.17 b	247.02
Urea + (NH_4_)_2_SO_4_	96.25 b	19.58 b	12.44 b	10.14 b	249.56
Urea + CaSO_4_	100.94 b	18.80 b	12.20 b	8.97 b	226.43
CaSO_4_·4urea	102.19 b	18.70 b	12.38 b	9.86 b	245.75
SE±	2.99	1.27	0.77	0.59	9.11
*p*-value	0.002	0.009	0.005	0.016	ns *
2019					
Control	83.25 a	16.63 a	10.07 a	8.50 a	265.00 a
Urea	88.00 a	17.35 ab	11.85 a	9.75 b	285.97 b
Urea + (NH_4_)_2_SO_4_	89.25 a	19.23 bc	11.82 a	10.50 bc	287.17 b
Urea + CaSO_4_	85.00 a	18.44 abc	11.87 a	10.77 c	285.76 b
CaSO_4_·4urea	97.00 b	20.17 c	14.53 b	12.00 d	286.02 b
SE±	2.13	0.77	0.59	0.31	4.66
*p*-value	0.006	0.044	0.003	0.000	0.024

Note. values in the same column followed by the same letter are not different (*p* < 0.05) according to Duncan’s multiple range test at the 5% level. * ns = not significant.

**Table 5 plants-11-03020-t005:** Analysis of variance for the quality of the grain listed in total N, crude protein (%), starch (%) contents, and grain yields, including a summary of correlation relationships between these variables and soil mineral N and sulfur content for the 2-year study (2018–2019).

Treatments	Grain N %		Crude Protein %		Starch %		Grain Yields	
2018								
Control	1.24 a		7.77 a		69.71		6.86 a 9.17 b 10.14 b 8.97 b 9.86 b	
Urea	1.34 b		8.39 b		68.73			
Urea + (NH_4_)_2_SO_4_	1.40 b		8.78 b		68.86			
Urea + CaSO_4_	1.38 b		8.62 b		67.60			
CaSO_4_·4urea	1.42 b		8.87 b		68.79			
*p*-value =	0.007		0.007		ns *		0.016	
2019								
Control	1.38 a		8.60 a		74.64 b		8.50 a 9.75 b 10.50 bc 10.77 c 12.00 d	
Urea	1.40 a		8.77 a		74.99 b			
Urea + (NH_4_)_2_SO_4_	1.43 ab		8.94 ab		73.41 b			
Urea + CaSO_4_	1.43 ab		8.93 ab		71.67 a			
CaSO_4_·4urea	1.47 b		9.22 b		70.16 a			
*p*-value =	0.049		0.056		0.000		0.000	
Correlation coefficient	*P* < *F*	*R*	*P* < *F*	*R*	*P* < *F*	*R*	*P* < *F*	*R*
2018								
Soil mineral N	<0.000	0.94	<0.000	0.93	ns *	−0.03	<0.000	0.74
Soil available sulfur	<0.000	0.79	<0.000	0.79	ns *	−0.07	<0.000	0.55
Total N in grains	-	-	<0.000	0.99	<0.000	−0.77	<0.000	0.81
2019								
Soil mineral N	<0.05	0.46	<0.05	0.47	ns *	−0.02	<0.000	0.62
Soil available sulfur	<0.000	0.61	<0.000	0.58	ns *	−0.07	<0.000	0.71
Total N in grains	-	-	<0.000	0.98	<0.000	−0.39	<0.000	0.76

Note. values in the same column followed by the same letter are not different (*p* < 0.05) according to Duncan’s multiple range test at the 5% level. * ns = not significant.

**Table 6 plants-11-03020-t006:** Analysis of variance for grain N uptake and ANR including a summary of relationships between these variables in 2 years (2018–2019).

Treatments	N Uptake in Grain (kg ha^−1^)				ANR (%) in Grain	
	2018		2019		2018	2019
Control	77.24 a		117.30 a		0.00 a	0.00 a
Urea	123.13 b		136.76 b		30.66 b	12.98 b
Urea + (NH_4_)_2_SO_4_	142.45 b		168.47 cd		43.44 b	34.12 cd
Urea + CaSO_4_	123.68 b		153.86 bc		30.94 b	24.38 bc
CaSO_4_·4urea	139.90 b		173.31 d		41.79 b	37.34 d
*p*-value	<0.000		<0.000		<0.000	<0.000
Correlation coefficient	*P* < *F*	*R*	*P* < *F*	*R*		
Soil mineral N	<0.000	0.79	<0.001	0.69	-	-
Soil available sulfur	<0.003	0.62	<0.000	0.73	-	-
Total N in grains	<0.000	0.87	<0.000	0.86	-	-
Grains yield	<0.000	0.99	<0.000	0.96	-	-

Note. values in the same column followed by the same letter are not different (*p* < 0.05) according to Duncan’s multiple range test at the 5% level.

## Data Availability

All data has been included in the main text.

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
