# Peer review of "Comparative Study Effect of Urea-Sulfur Fertilizers on Nitrogen Uptake and Maize Productivity"

_plants, 2022, doi:10.3390/plants11223020_

Round 1

Reviewer 1 Report

Well designed research, appropriately written, minor corrections needed

Author Response

Thanks so much for your comments. It was useful and supportive.

Reviewer 2 Report

The manuscript entitles "Comparative Study Effect of Urea-Sulfur Fertilizers on Nitrogen Uptake and Maize Productivity" is an interesting investigation but in this current form the manuscript have several technical flaws not only in write up but also in the methodology. Also the presentation of the results is very confusing which makes difficult to understand the core findings. There have been several investigations in the use of Sulfur urea fertilization and significant results have been found. I feel the novelty of the manuscript at this stage is to investigate the benefit cost ratios of the most effective product which is missing in the manuscript. Several other issues which makes this manuscript not acceptable at this stage are as follows.

1. Abstract is very qualitative instead of normal summary of quantitative date representation.

2. Introduction is very short with very old irrelevant references for example line 31 reference 1 also in the introduction write  up is very poor in the line 31 year is 198 i think it is 1980. There are several new investigations on coated urea with various material why author choose to start very old method in introduction. Understanding sentences is very difficult please check first paragraph with very long sentence.

3. Material and methods have technical flaws for example the line 69 shows the depth of sampling only 0-20 cm. Normal sampling for Maize crop is 0-30cm why authors took sample not properly according to standard methods which is 0-15 and 15-30. Also planting geometry is strange.

Result section need intensive revision.

conclusions are very qualitative.

references are not formatted according to journal style

Author Response

(The authors gave the same response as above.)

Reviewer 3 Report

The manuscript “Comparative Study Effect of Urea-Sulfur Fertilizers on Nitrogen Uptake and Maize Productivity” is a very interesting contribution and deserves to be published in Plants. In this way, the authors should carefully review the format of the journal and edit the format and grammar mistakes that were seen in whole manuscript. In addition, some sections of material and method sections should be deeply explained in relation to the results. The discussion and conclusion section should be remade according to the findings.

Other comments:

L8-24: Please, review guide of authors to place background, methodology, results and conclusions of the abstract.

L10-11: Why combining N+S it is possible to reduce loss of N by volatilization?

L17: I suggest to remove “by p>.016 and p>.001 for 2018 and 2019, respectively”.

L20-22: Please, improve grammar and text of this sentence for more understanding.

L31: What’s means “since 198”?

L33: Soil and other words in whole manuscript should be written in lowercase.

L49-51: Not only in cereals, but other crops also improved their productivity and quality based onto N+S applications. Please, review the following references:

1.     https://doi.org/10.1016/j.fcr.2018.07.010

2.     https://doi.org/10.1016/j.lwt.2016.08.039

3.     https://doi.org/10.1016/j.jcs.2020.102969

L53-57: Some authors suggest that sulphur decrease leaf or tissues pH, inducing a higher concentration gradient. Please, add some information about this.

L57-63: This sentence is written in another format, please, review it.

L77-83: This information should be placed in a continuous form.

L91-92: I suggest to add little information about climate since the maize was harvested in a different date at the same physiological maturity. Or to add climate information before this sentence.

L104-105: I am not sure if it is possible to start a phrase using while. Please, review it.

L106-109: I suggest to add the reference in which the laboratory bases the analysis of mineral N.

L118-127: Please, follow the guide of the authors to describe text of equations.

L163: Please, replace units “mg/kg-1” by “mg kg-1”.

L183-191: Please, remade the figure adding the correct units. In addition, the A,B,C,D and significant differences should be written in a continuous form.

L192-224: These paragraphs are written in another format, please, review it.

L226-223: Please, remade the figure adding the correct units. In addition, the A,B,C,D and significant differences should be written in a continuous form.

L246-249: I suggest to add standard deviation of each obtained mean.

L251-269: I suggest to compare the treatments in this section. In most of them, the authors should add “higher than”.

L270-273: I suggest to add standard deviation of each obtained mean.

L270-278: Tables and figures should be placed after text explanation.

L283-287: I suggest to add standard deviation of each obtained mean.

L324-327: I suggest to add standard deviation of each obtained mean.

L348-355: Discussion should not be placed with sections.

L353-391: Most of the information placed in this section is stated in results section. I suggest to remove most of the repeated information of the results section and do not repeat it.

L410-420: Most of the information are results and not discussion. I suggest to edit or remove it.

L410-423: I suggest to discuses the results based onto the findings on starch.

L425-433: If the authors compared treatments, it should be difines which treatments reached the best performance in productivity. This information should be highlighted in all the sections, including abstract, discussion and conclusions.

L434: Please, remove the comment performed in acknowledgments.

L451-644: I suggest to use Mendeley for the references.

Page 21 to 24 should removed.

Author Response

Thanks so much for your comments. It was useful and beneficial.

Round 2

Reviewer 2 Report

The manuscript is revised according to suggestion and can be accepted for publication in this current form.

Author Response

Thank you so much for your comment and your cooperation!

It was helpful and encouraged me to improve my manuscript.

Reviewer 3 Report

Despite that the authors have made a great effort to improve the manuscript, they only edited the text, grammar and english, but not the substantive issues. The authors should reread the review again the first revision and correct what they didn't want to review previeously. I refers to add more references and to discusse more in relation to other components. On the other hand, mistakes in relation to units in the figures were not edited. In addition, Figure 4 is should contain similar letter, not upper and lower case (this is usually when compared seasons).

Author Response

  • Despite that the authors have made a great effort to improve the manuscript, they only edited the text, grammar and english, but not the substantive issues. The authors should reread the review again the first revision and correct what they didn't want to review previously.
    Thanks for your comments but I tried to do my best to revise my manuscript according to your comments as much as I can I hope I did it successfully this time.
  • I refer to adding more references and discussing more in relation to other components. I added more references.
  • On the other hand, mistakes in relation to units in the figures were not edited. I edited it.
  • In addition, Figure 4 is should contain similar letter, not upper and lower case (this is usually when compared seasons).
  • There is no significant difference between both seasons when means are compared so I tried to mention the differences between the treatments for every season separately in lower and upper case.